# Self-implantable double-layered micro-drug-reservoirs for efficient and controlled ocular drug delivery

Aung Than[1], Chenghao Liu[2], Hao Chang[1], Phan Khanh Duong[1], Chui Ming Gemmy Cheung [3,4,5], Chenjie Xu[1], Xiaomeng Wang[2,3,6,7] & Peng Chen[1]

Eye diseases and injuries impose a significant clinical problem worldwide. Safe and effective ocular drug delivery is, however, challenging due to the presence of ocular barriers. Here we report a strategy using an eye patch equipped with an array of detachable microneedles. These microneedles can penetrate the ocular surface tissue, and serve as implanted micro-reservoirs for controlled drug delivery. The biphasic drug release kinetics enabled by the double-layered micro-reservoirs largely enhances therapeutic efficacy. Using corneal neo-vascularization as the disease model, we show that delivery of an anti-angiogenic monoclonal antibody (DC101) by such eye patch produces ~90% reduction of neovascular area. Furthermore, quick release of an anti-inflammatory compound (diclofenac) followed by a sustained release of DC101 provides synergistic therapeutic outcome. The eye patch application is easy and minimally invasive to ensure good patient compliance. Such intraocular drug delivery strategy promises effective home-based treatment of many eye diseases.

[1] School of Chemical and Biomedical Engineering, Nanyang Technological University, 70 Nanyang Drive, Singapore 637457, Singapore. [2] Lee Kong Chian School of Medicine, Nanyang Technological University, 59 Nanyang Drive, Singapore 636921, Singapore. [3] Singapore National Eye Centre, 11 Third Hospital Avenue, Singapore 168751, Singapore. [4] Singapore Eye Research Institute, 20 College Road, Singapore 169856, Singapore. [5] Department of Ophthalmology, Yong Loo Lin School of Medicine, National University of Singapore, 1E Kent Ridge Road, NUHS Tower Block Level 7, Singapore 119228, Singapore. [6] Institute of Molecular and Cell Biology, Agency for Science, Technology & Research, 61 Biopolis Drive, Proteos, Singapore 138673, Singapore. [7] Institute of Ophthalmology, University College London, London EC1V 9EL, UK. Correspondence and requests for materials should be addressed to C.X. (email: cjxu@ntu.edu.sg) or to X.W. (email: WangXiaomeng@ntu.edu.sg) or to P.C. (email: chenpeng@ntu.edu.sg)

The increasing prevalence of eye diseases (e.g. glaucoma, diabetic retinopathy, age-related macular degeneration, etc.) is correlated to the upsurge in aging population, diabetes mellitus and prolonged wear of contact lens worldwide[1–3]. However, efficient delivery of drugs into the eye is challenging due to the presence of multiple structural barriers (e.g. corneal epithelium and blood–retinal barrier)[4,5]. The use of systemic route (parenteral or oral administration) requires a large dose to achieve effective local drug concentration, and thus usually produce off-target systemic side effects[6,7]. On the other hand, repetitive drug applications with high dosage are often required for convention topical administration (e.g. eye drops or ointments) due to extremely low bioavailability (<5% can be absorbed by eye) and fast clearance, which may also lead to systemic side-effects (e.g. prolonged steroid eye drop usage causes not only ocular hypertension but also systemic toxicity like uncontrolled hyperglycaemia)[6–8]. Intraocular injection (e.g. intracameral and intravitreal injection) using conventional hypodermic needles to penetrate the surface barriers (cornea and sclera), however, has poor patient compliance due to pain, need for frequent clinic visit, risk of infection, haemorrhage, even permanent damage[9]. Similar to topical eye drops, injecting drugs into ocular surface tissues (e.g. corneal intrastromal layer, sclera) also has poor drug retention due to back-flow of injected solution and subsequent tear wash-out[4,6]. Furthermore, both conventional topical administration and local injection only produce burst release of drug with short effective duration, which is particularly not ideal for treating chronic progressive eye diseases, such as glaucoma[6,10]. Although contact lens-like hydrogels have been developed for improved topical delivery, because of prolonged drug residence time with minimal burst effect[11,12], the bioavailability is still poor. Although implanting intraocular drug reservoirs enables sustained release, it requires risky and painful surgical intervention[13]. Hence, localized, long-lasting and efficient ocular drug delivery with good patient compliance is still an unmet medical need.

Microneedle (MN) technology is originally developed for transdermal drug delivery for various therapeutic purposes (e.g. vaccination, local anaesthesia, anti-diabetic and anti-obesity treatments), with painless, bloodless, high efficiency and ease of administration properties[14–16]. Their patient-friendly feature and effectiveness in transdermal drug release have inspired researchers and clinicians to explore their applications in eye disease treatment. Specifically, drug-coated solid stainless-steel MN have been used for the rapid release of drugs in the cornea[17,18] and hollow glass MN have been employed to infuse drug solution into the sclera[19].

Here, we show a flexible polymeric eye patch equipped with an array of biodegradable and detachable MNs for localized, highly efficient and controlled ocular drug delivery (Fig. 1). MNs can penetrate the ocular barriers (epithelial and stromal layers of the cornea) with minimal invasiveness and be self-implanted as drug reservoirs for controlled drug release. The double-layer structured MNs allow biphasic release kinetics and packaging of multiple drugs for synergistic therapy. As the proof-of-concept demonstration, we show the superior effectiveness of such eye patch in the treatment of corneal neovascularization (NV) as compared to topical eye drop and fast drug-release approaches. A swellable eye patch without MNs is also used to collect eye fluid for monitoring the therapeutic effectiveness based on biomarker detection. We believe this approach could be paradigm-shifting for long-term home-based treatment and management of various eye diseases.

## Results

**Fabrication of eye patch with double-layered microneedles.** Hyaluronic acid (HA) is a non-sulphated glycosaminoglycan distributed abundantly throughout the body in the connective tissues as well as vitreous eye fluid. As a natural biopolymer with unique viscoelastic property and transparency, HA has been widely used in ophthalmology, particularly in artificial tear solution as a lubricant for dry eyes[20]. HA-based MN devices have been employed for transdermal delivery of various hydrophilic or hydrophobic therapeutic compounds, including proteins, peptides and synthetic molecules[14–16]. However, because of the fast dissolving nature of HA, HA-MNs cannot maintain their sharp-pointed structural integrity and mechanical strength during penetration into a wet surface like cornea. In addition, HA-MN can only afford burst release of its cargo[14]. In comparison, crosslinked methacrylated HA (MeHA), which is synthesized by functionalizing HA with methacrylic anhydride (Supplementary Fig. 1) is more resistive to dissolution and offers a slow release of its cargo[21,22]. But the stiffness of MeHA-MNs is inferior to HA-MNs[23,24].

Combining the merits of HA and MeHA, we herein developed an eye-contact patch equipped with double-layered MNs (DL-MN) for controlled ocular drug delivery (Fig. 1), using a simple micro-moulding method (Fig. 2). The MNs have a HA inner core and a MeHA outer layer. Because the highly dissolvable HA is covered by MeHA, the MNs are able to penetrate the wet cornea surface. Briefly, a small amount of MeHA aqueous solution, with or without therapeutic compounds, was centrifuged into the reverse MN structures in the female polydimethylsiloxane (PDMS) mould. Hollow MN structures were formed after drying in ambient overnight as hydrophilic and viscous MeHA polymers tend to stick onto the hydrophilic surface of MN cavities in plasma-treated PDMS mould. Subsequently, unmodified HA solution, with or without therapeutic compounds, was filled in the remaining cavities and air-dried to form solid MNs. Finally, pure HA solution was introduced into the PDMS mould on top of the MN array to make the supporting substrate. After drying, the MN patch was peeled off from the mould and subject to a brief exposure to ultraviolet light to crosslink MeHA outer layer of MNs.

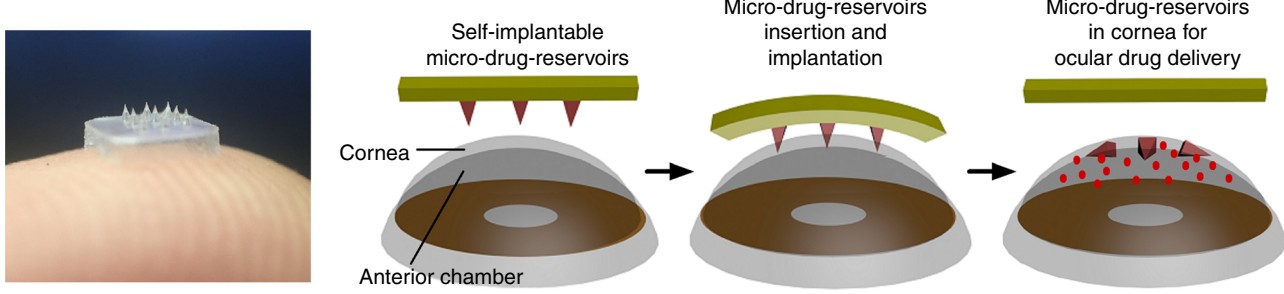

Self-implantable micro-drug-reservoirs

Micro-drug-reservoirs insertion and implantation

Micro-drug-reservoirs in cornea for ocular drug delivery

Cornea

Anterior chamber

**Fig. 1** Illustration of eye-contact patch for ocular drug delivery. The eye patch is equipped with an array of self-implantable micro-drug-reservoirs

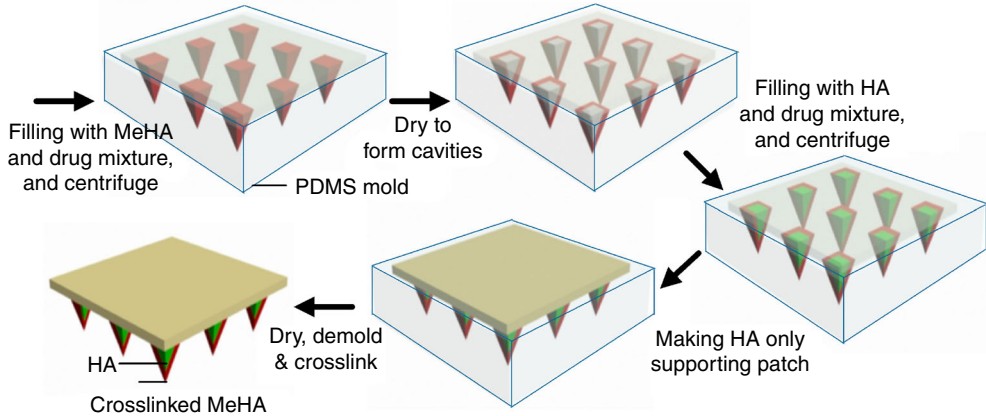

**Fig. 2** Schematic illustration. Fabrication process of the polymeric patch with an array of needle-shaped and double-layered micro-drug-reservoirs

As revealed by the scanning electron microscopy (SEM) and optical microscopy (Supplementary Fig. 2), the fabricated patch consists of an array of pyramidal-shaped MNs with tip diameter of ~10 μm, height of ~500 μm, base width of ~250 μm and inter-needle spacing of ~400 μm. The final MN dimensions are smaller than the stainless steel templates (300 μm bases with 600 μm height), due to the shrinkage of PDMS and HA/MeHA during the fabrication process. The MN design is based on the previous findings that pyramidal-pointed tips (compared to conical one) with the aspect ratio of 2:1 (height to base diameter) are optimal for tissue penetration[16,25]. The eye patches (~2 × 2 mm) with a 3 × 3 MN array were used for mice whose cornea size is ~3 mm in diameter (Supplementary Fig. 2).

Immunoglobulins (IgG) labelled with Alexa Fluor 680 or Alexa Fluor 488 as the model therapeutic compounds were separately loaded in the different layers of MNs (Fig. 3a, b). The confocal fluorescence imaging confirms that red IgG(680) and green IgG (488) can be separately encapsulated in the outer and inner layers of MN, respectively, while the substrate is free of IgG molecules (Fig. 3c–e). Upon insertion of MNs into the tissue simply by thumb pressing on the supporting substrate, the tissue fluid is drawn into the MNs and quickly dissolves the interfacial HA layer between the MNs and substrate whereby causing detachment of MNs (Fig. 1). The embedded DL-MNs serve as the micro-reservoirs for localized and sustained drug release. The inner HA core being exposed to the tissue fluid undertakes quick dissolution and discharge of the cargo, whereas the outer MeHA layer dissolves slowly letting the cargo molecules slowly seep through the crosslinked polymer matrix.

IgG recovered from 3 × 3 DL-MNs dissolved in phosphate buffer saline (PBS) is 0.92 ± 0.21 μg, correlating well with the nominal loading amount (1 μg) (Supplementary Fig. 3a). In vitro stability of encapsulated IgGs was tested by evaluating their molecular weight on polyacrylamide gel electrophoresis after storage of MN patch at 4 °C for 1 week. Majority of IgGs (82.11 ± 11.2%) released from either HA or crosslinked MeHA matrix was intact as evidenced by the expected band at ~150 kDa (Supplementary Fig. 3b). The bioactivity of IgGs (specific to vascular endothelial growth factor receptor 2, VEGFR2) released from MNs (being stored for 5 days) over 24 h duration was confirmed by immunofluorescence staining demonstrating the capability of binding and recognizing VEGFR2 on endothelial cells (Supplementary Fig. 3c). The bioactivity of IgGs released from MNs (being stored for 5 days) over 6, 24 or 120 h duration was further confirmed by their inhibitory effect on endothelial cell tube formation (Supplementary Fig. 4a). Furthermore, in vitro biocompatibility of DL-MNs was proven by the well-preserved morphology and viability of corneal epithelial cells with

the presence of either un-modified HA or crosslinked MeHA (Supplementary Fig. 4b).

**Characterization of double-layered micro-drug-reservoirs**. In vitro drug release kinetics was examined by monitoring the release profiles of different IgG molecules encapsulated in the two compartments of DL-MNs. As shown in Fig. 3f, IgG(488) loaded in the fast dissolving inner HA layer can be quickly released in both artificial tear fluid (which mimics tear) and gelatin hydrogel (which mimics corneal stromal tissue[26]). Specifically, >80% was released within 5 min in the former and 30 min in the latter. In contrast, prolonged release profile was observed for IgG(680) loaded in the crosslinked MeHA outer layer ($t_{1/2}$ of ~2 days in tear fluid and ~3 days in gelatin hydrogel) because IgG molecules can only slowly diffuse through the interwoven meshwork of MeHA. In comparison to the fast release from HA-MNs or slow release from MeHA-MNs, the bi-phasic release profile was realized when a single drug molecule, IgG(680), was loaded in both compartments of DL-MNs (Fig. 3g).

To predict in vivo drug release profile, DL-MNs were embedded within the agarose hydrogel (which mimics corneal tissue in which water content is ~80%) and were continuously monitored under confocal microscope (Fig. 4a). Immediately after insertion, DL-MNs were quickly detached from the supporting substrate (<60 s) into the hydrogel because fluid was quickly drawn into MN-substrate junction. As the supporting substrate is made of highly-dissolvable low-molecular-weight HA (3–10 kDa), HA molecules at the MN-substrate junction dissolve rapidly as the fluid inside and at the surface are quickly drawn into the hydrophilic HA matrix. The patches were then removed allowing MNs embedded into the hydrogel. The mechanic perturbation caused by the penetration and substrate removal process also facilitates MN detachment. The fast-dissolving HA inner-core released the loaded green IgG(488) within 10 min, while the crosslinked MeHA outer-shell gradually swelled and slowly discharged the encapsulated red IgG(680) into the hydrogel (Fig. 4b, Supplementary Fig. 5a). The real-time release profile of double-layered micro-implants in hydrogel was captured by fluorescence microscopy as shown in Fig. 4b. Taken together, our data demonstrate the biphasic release kinetics of DL-MNs, i.e. a burst phase followed by a slow discharge over several days. Note that MNs in hydrogel were barely visible under bright field imaging (Supplementary Fig. 5), suggesting that they are essentially transparent and hence suitable for use in corneal tissue.

The mechanical strength of MNs was assessed by compression test. Consistent with other studies, the mechanical strength of

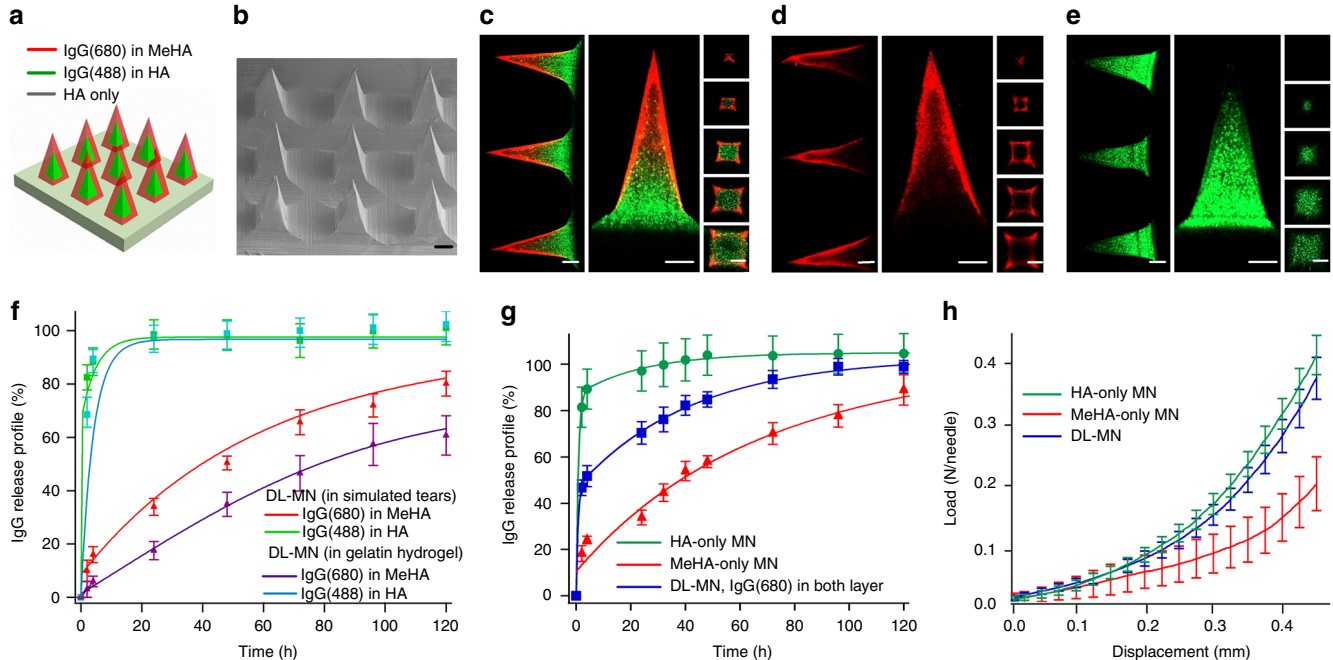

**Fig. 3** Characterization of double-layered microneedles (DL-MNs). **a**, **b** Schematic and SEM images of a polymeric patch with an array of DL-MNs (the outer layer made of crosslinked MeHA and inner core made of HA). Scale bar = 100 μm. **c–e** Representative confocal images of **c** DL-MNs loaded with immunoglobulin G conjugated with Alexa Fluor: IgG(680) (red colour) in outer layer and IgG(488) (green colour) in inner core, **d** DL-MNs with IgG(680) in outer layer only, **e** DL-MN with IgG(488) in inner core only. Scale bars = 100 μm. **f** In vitro fast and slow release profiles of DL-MNs in simulated tears or gelatin hydrogel (37 °C). IgG(680) was loaded in outer layer and IgG(488) was loaded in inner core (n = 3). **g** In vitro release profiles of IgG(680) from HA-MN, MeHA-MN or DL-MN in phosphate buffer saline (PBS) (37 °C) (n = 3). IgG(680) was loaded in both layers of DL-MN. **h** Mechanical compression test of HA-MN, MeHA-MN and DL-MN (n = 4). The data represents as mean value, and error bars indicate SEM (mean ± SEM). n represents the number of samples for each group

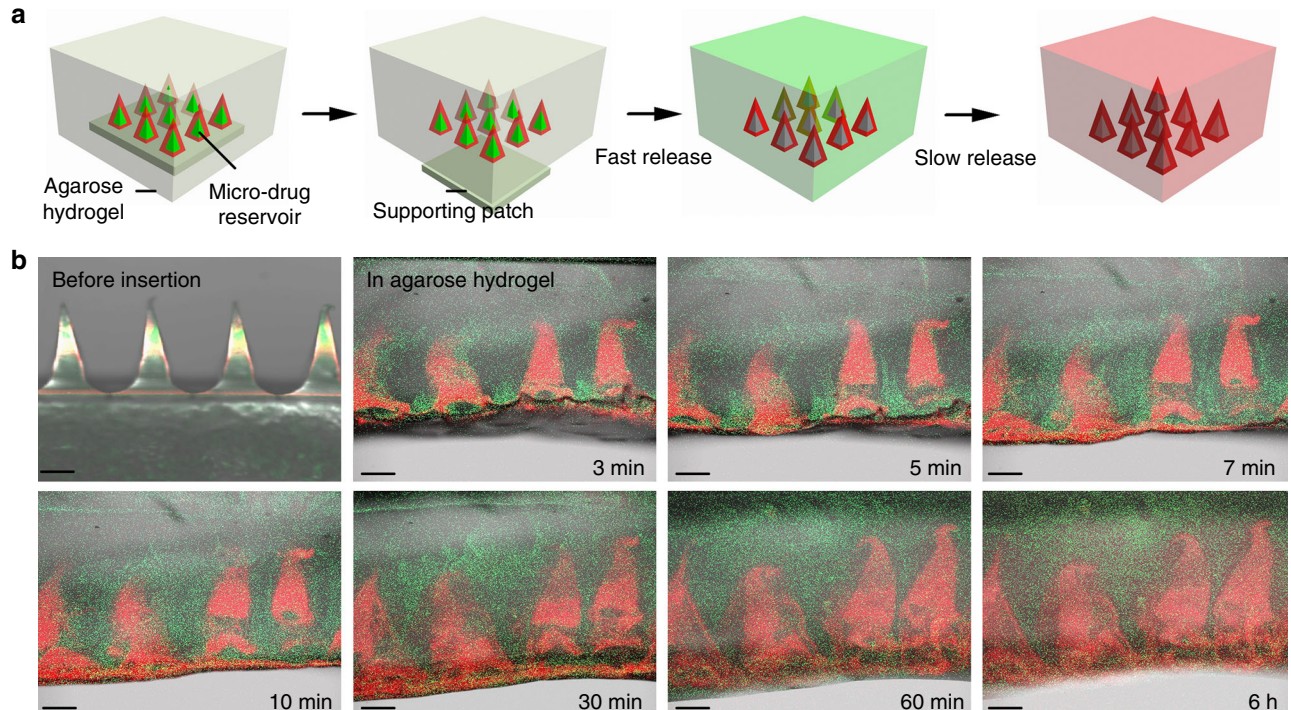

**Fig. 4** Biphasic release profile of double-layered microneedles. **a** Schematic release profile of DL-MNs in agarose hydrogel. **b** The merge of the optical and fluorescence image of DL-MNs with the supporting patch (before insertion), and visualization of real-time release of IgG(680) (red colour) and IgG(488) (green colour) from DL-MNs in agarose hydrogel (3 min–6 h). Scale bars = 200 μm

HA-MNs (~0.4 N per needle) is strong enough for skin penetration (Fig. 3h)[27]. In contrast, the mechanical strength of crosslinked MeHA-MNs (~0.15 N per needle) is much weaker. Because MeHA is highly viscous, only a lower polymer concentration can be used to fabricate MNs (50 mg/ml for MeHA-MNs vs. 200 mg/ml for HA-MNs). This compromises the mechanical strength of MeHA. The mechanical strength of DL-MNs is similar to that of HA-MNs (~0.4 N per needle), indicating that the mechanic property of DL-MN is dictated by the inner HA core. In addition, drug loading into DL-MNs (2 µg) does not compromise the mechanical properties of DL-MNs (Supplementary Fig. 2g). It has been reported that, when an 18G hypodermic needle (outer diameter of 1.27 mm) is used, the force required to penetrate human cornea is ~0.5 N per needle[28]. Considering the sharp tip of a MN (~10 µm), the force needed to penetrate the cornea can be much lower[29].

To further confirm the insertion capability and drug releases of DL-MNs, MN-patches were applied on the porcine cornea simply by thumb pressing (~1.9 N, ~30 s) (Fig. 5a). Porcine eye has been commonly used as a good model for studying cornea, as its anatomical structures, water content and thickness (~0.9 mm) are similar to that of human cornea (~0.6 mm)[30]. As demonstrated in Fig. 5c, d, MNs were well-penetrated and embedded into the porcine cornea. The subsequent histological study shows cavities into the corneal stromal layer ~150 µm deep which is about one-third of MN height (Fig. 5e). This is because of compressive deformation of MN (Supplementary Fig. 2h), and the elastic property of the cornea provided by the densely-packed intertwining collagen fibrils[31]. Such elastic deformations make polymeric DL-MNs advantageous over metallic MNs which may cause corneal puncture[32]. The insertion force into the cornea was estimated (Fig. 5f). The force required to penetrate is ~0.05 N per needle, as indicated by the transition point where resistance from the tissue sharply increases. The success rate of MN embedment is ~85% as evidenced by visual inspection of the removed substrate (Fig. 5c, d) and the fluorescence marks left behind (Fig. 5g). The fluorescence spots in the cornea from the embedded MNs were further analysed by confocal microscopy

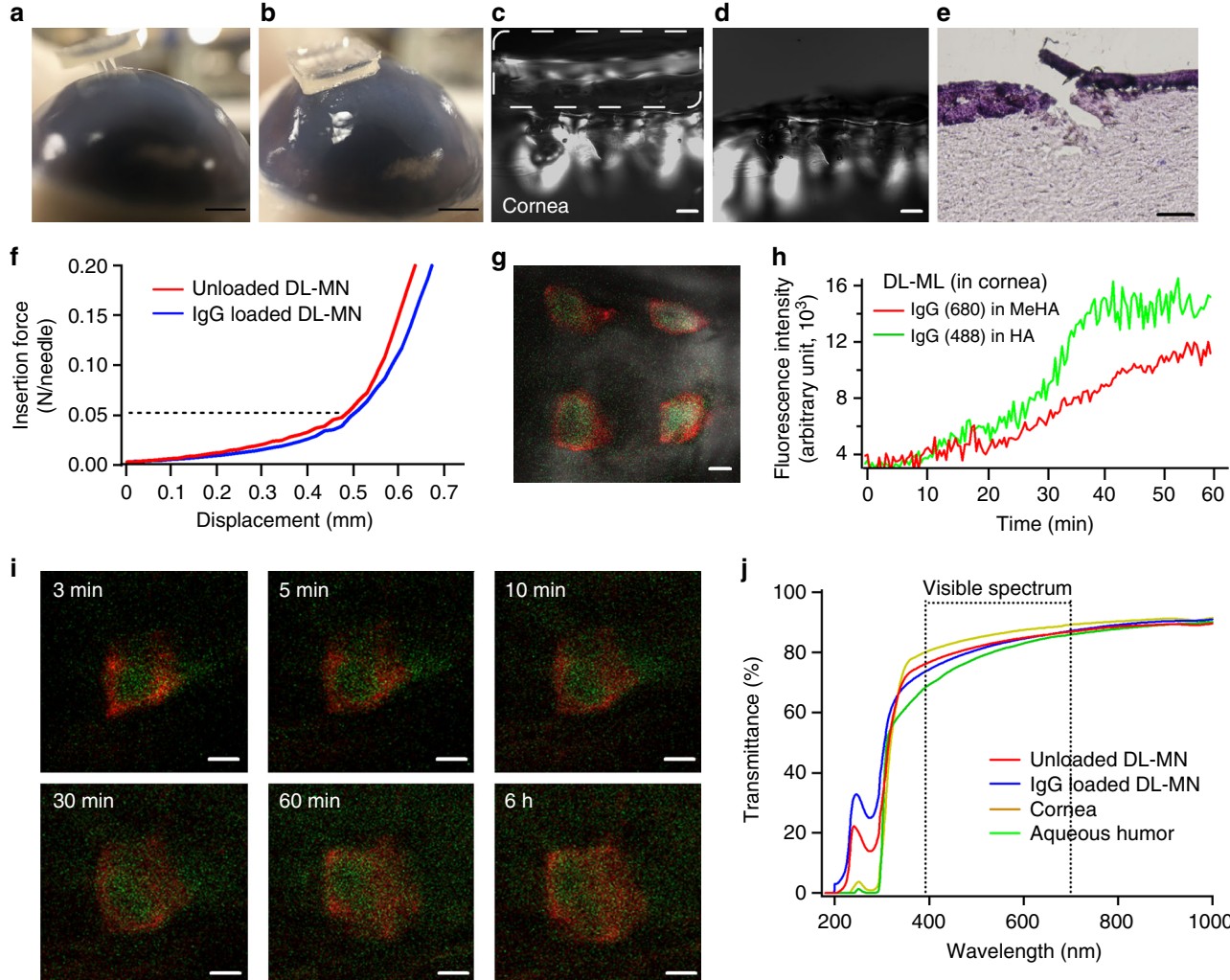

**Fig. 5** Ex vivo studies of double-layered microneedles on porcine cornea. A polymeric patch containing 3 × 3 DL-MNs was applied on the central region of porcine eye (~30 s). **a**, **b** Bright-field images of the cornea before (**a**) and after (**b**) MN insertion. Scale bars = 2 mm. **c**, **d** Bright-field images of the cornea (cross-sectional view), upon MN insertion (**c**) and after removal of the supporting patch (**d**). Scale bars = 200 µm. **e** Hematoxylin and eosin stained section of the cornea showing the cavity caused by DL-MN penetration. Scale bar = 100 µm. **f** The averaged insertion force of DL-MN into the cornea (*n* = 3). **g–i** Confocal image of embedded DL-MNs in cornea (**g**), the average fluorescence intensity changes (*n* = 4) in the region adjacent to DL-MNs (**h**), and confocal visualization of real-time release of IgG(680) (red colour) and IgG(488) (green colour) from DL-MN into the cornea (**i**). Scale bars = 100 µm. **j** Transmittance of the fully-hydrated DL-MN, cornea and aqueous humour at different wavelengths (*n* = 3). *n* represents the number of samples for each group

(Fig. 5i). Similar to the results shown in Fig. 4, HA inner-core quickly discharged its cargo IgG(488) while the crosslinked MeHA outer-shell slowly released IgG(680) into the corneal tissue. We also observed that the transmittance in the visible range of fully hydrated DL-MN is about 73–86%, which is comparable to that of cornea and aqueous humour (Fig. 5j)[33]. Taken together, these experiments demonstrate that DL-MNs are strong enough to penetrate into the cornea, easy to detach from the supporting patch after insertion, transparent inside the cornea and capable of biphasic release kinetics.

**In vivo studies of biosafety and ocular drug delivery**. The capability of insertion and drug releases of DL-MNs was further investigated in vivo. DL-MN patch loaded with IgG(680) was gently pressed onto the cornea of mouse eye for ~30 s (Fig. 6a). MNs were quickly detached from the supporting substrate and implanted into the cornea. MN patch before and after being applied onto the cornea is shown in Fig. 6b and the detachment of MNs is evidenced. The treated eyes were flushed with PBS, and then examined with the fluorescence and bright-field imaging. In vivo fluorescence images indicate that fluorescence intensity of IgG(680) was strong only at the MN insertion spots, but absent in the control eye (Fig. 6c), suggesting the successful implantation of MNs into the cornea. Figure 6d shows a representative bright-field image of the mouse eye immediately after MN insertion (day

0). The MN applied corneas were further analysed under confocal microscope. It was found that IgG(680)-loaded MNs were embedded within the cornea (~90% success rate), as evidenced by visual inspection of the removed substrate (Fig. 6b) and the fluorescence marks left behind (Fig. 6e). Subsequent histological examination revealed the small penetration cavities (~100 μm) inside the corneal stromal layer, similar to the observation in porcine cornea (Fig. 5e).

The corneal anatomical structures after MN insertion were monitored over 1 week (Fig. 6f). We did not observe any puncture on cornea in all experiments (Fig. 6f), suggesting that DL-MNs are strong enough to penetrate into the stromal layer, but not too stiff to spear throughout the whole cornea. And the small insertion marks were only observed immediately after insertion in all mice (day 0) and almost disappeared after 24 h (Fig. 6f), suggesting the closure of epithelium through the spontaneous repairing process. At days 3 and 7, DL-MN-treated corneal epithelium restored its structural integrity and appeared as normal as untreated cornea in all mice (Fig. 6f). There were no significant differences of body weight and food intake between the control and test groups (Fig. 6g, h). And there were no visible indications of corneal opacity, inflammation or haemorrhage in any MN-treated eyes (Fig. 6d). We also did not observe any signs of pain in the test group based on the grimace scale pain assessment (Fig. 6i)[34]. All these observations indicate that MN insertion and implantation into cornea is minimally invasive

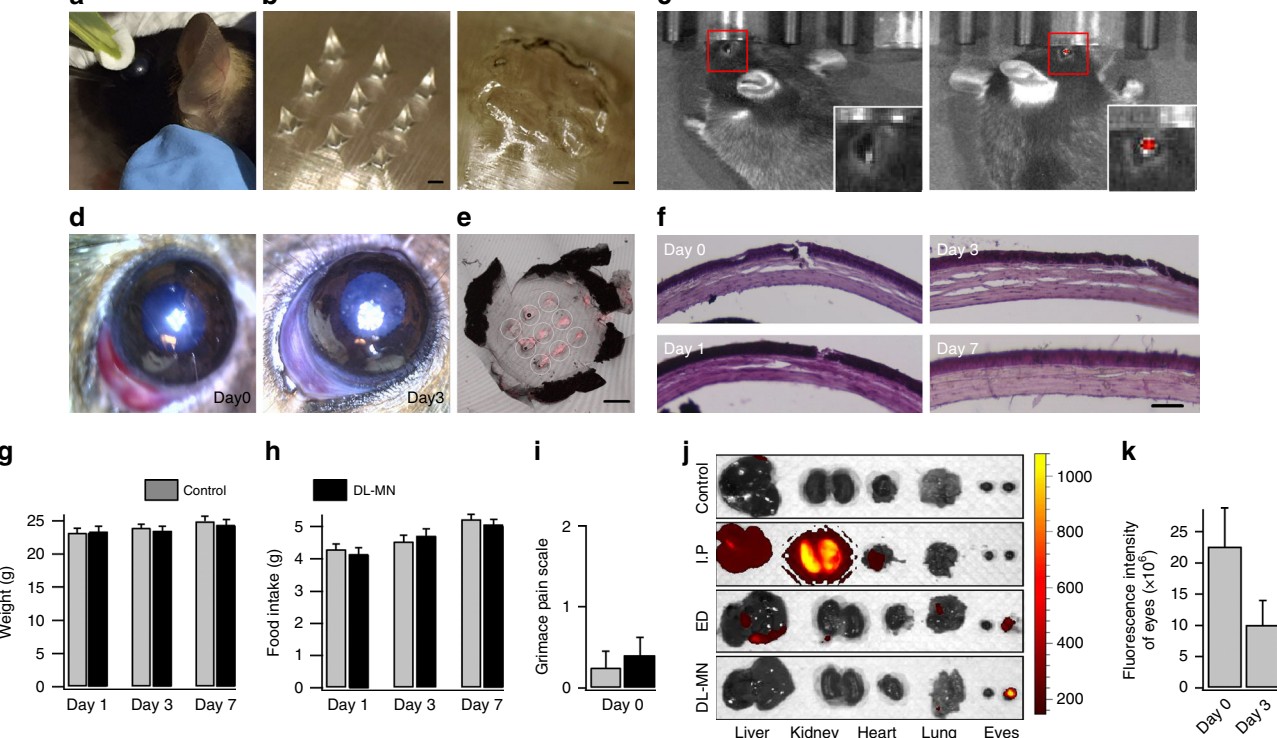

**Fig. 6** In vivo studies of self-implantable double-layered microneedles. **a** A polymeric patch containing 3 × 3 DL-MNs was applied on the central region of mouse eye (~30 s). **b** Bright-field images of the patch, before (left) and after (right) insertion into the eye, are shown. Scale bars = 200 μm. **c** In vivo imaging of the eyes, applied without (left) or with IgG(680)-loaded MN patch (right). **d, e** Bright field images of the eye treated with a IgG(680)-loaded MN patch, at day 0 (immediately after insertion) and at day 3 (**d**). The red fluorescence spots in the cornea mark the penetration sites (**e**). Scale bar = 500 μm. **f** The representative histological changes of mouse cornea, at day 0, day 1, day 3 or day 7 after MN patch application. Scale bar = 100 μm. **g–i** The weight (**g**), food intake (**h**), and the grimace scale for pain assessment (**i**) were examined at day 0 (2 h after insertion), day 1, day 3 or day 7 after patch application. Data represent mean ± SEM (n = 5). **j, k** In vivo distribution of IgG in mice, treated without (control) or with systemic injection (intraperitoneal, I.P.), eye drop (ED) (one-side only) or MN patch application (one side only) of IgG(680). The representative fluorescence images of dissected organs, 2 h after treatment, are shown in (**j**). The quantitative analysis (mean ± SEM; n = 5) of fluorescent intensities of eyes treated with MN patch at day 0 and day 3 is shown in (**k**). n represents the number of samples for each group

without causing obvious adverse effects on the eyes, general health state and behaviour.

As shown in Fig. 6j, systemic injection (peritoneal injection) of 10 μg IgG(680) led to strong fluorescence in all major organs (liver, kidney, heart, lung, etc.), but not the eyes. Local instillation of IgG(680) containing eye-drop also produced fluorescence signal in liver and lung, in addition to the treated eye. This is consistent with the notion that topical instillation results in systemic drug distribution away from the eye via the highly vascularized conjunctiva[7]. In contrast, intra-corneal delivery of IgG(680) at the same dose using MN patch only introduced fluorescence signal in the applied eye (much stronger than that caused by eye-drop). In addition, fluorescence signal in the eye was still observable for over 3 days (Fig. 6k), showing that the intra-corneal micro-implants act as drug-depots for localized and sustained ocular drug delivery.

**Double-layered MNs improve the efficacy of anti-VEGF therapy.** Eye trauma, including chemical injury and infection, can trigger corneal NV, and may cause corneal opacity, visual

impairment and even blindness. Studies have shown that vascular endothelial growth factor (VEGF) is a key mediator of corneal NV. VEGF promotes blood vessel formation mainly via VEGF receptor type 2 (VEGFR2)[35]. Recently, anti-VEGF therapies (e.g. ranibizumab) have become standard care for vaso-proliferative diseases, including those in the eye[36–38]. However for angiogenic ocular diseases like corneal NV, frequent high-dose topical application is required to overcome ocular barrier (e.g. corneal epithelium), which is accompanied with adverse effects (e.g. subconjunctival haemorrhage)[8,37]. On the other hand, conventional intraocular injection may cause infection, bleeding and retinal detachment[9]. Here, we demonstrate the advantages of DL-MN eye-patches for improving the therapeutic efficacy of the anti-VEGF drug (Fig. 7).

Corneal NV was induced by the well-established alkali burn-injury model (day 0). At day 2 when neovascular growth was started to appear from the limbus (the border between cornea and sclera), mice corneas were treated once with non-specific control IgG or anti-VEGFR2 IgG (DC101) delivered through topical eye-drop or MN patch application (Fig. 7). DC101 was used because of its proven effectiveness against angiogenesis in murine

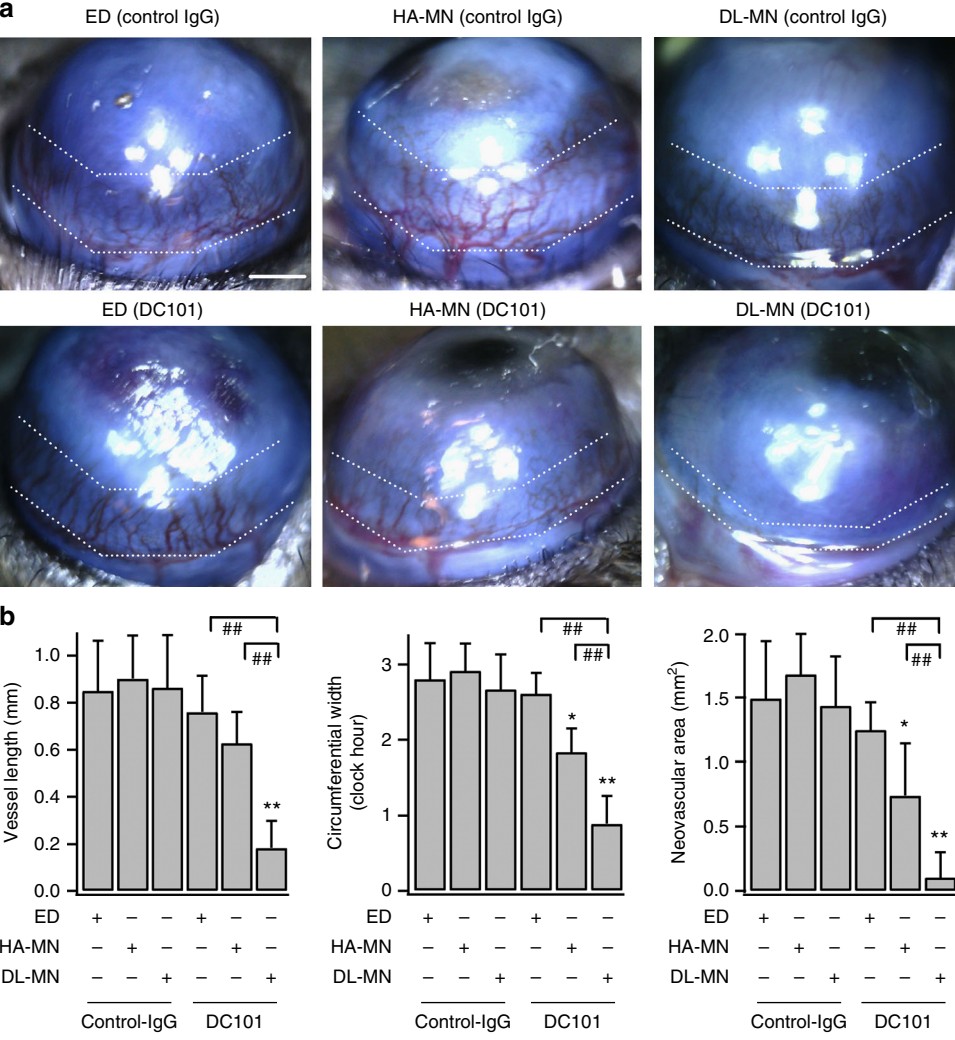

**Fig. 7** DL-MN patch improves the therapeutic efficacy of anti-VEGF therapy. Mouse eyes were treated differently 2 days after being inflicted with alkali-burn, and examined at day 7. **a** Representative images of differently treated eyes and **b** quantifications of corneal neovascularization (mean ± SEM; 6 samples for each group). Scale bar = 500 μm. The white dotted lines indicate the extent of neovascular outgrowth from the limbus. Statistical comparison between groups was performed using one way ANOVA. *$p < 0.05$, **$p < 0.01$ vs. control (control IgG eye drop, ED); ##$p < 0.01$ between indicated pairs

models[39]. Similar to the un-treated eyes, eyes treated with control IgG eye-drop showed substantial corneal NV ($1.48 \pm 0.45$ vs. $1.50 \pm 0.24$ mm$^2$) (day 7). Similarly, topical delivery of DC101 via eye-drop (with a high dose of 10 µg in 10 µl) had no significant effect on corneal NV as compared to the untreated eyes ($1.24 \pm 0.21$ mm$^2$). In contrast, eyes treated with DC101 (~1 µg) delivered through fast-dissolving HA-only MNs led to ~44% reduction in neovascular area ($0.84 \pm 0.43$ mm$^2$) (Fig. 7). This is consistent with the previous studies that ~40% reduction of corneal NV can be expected using rapid ocular delivery approaches (e.g. subconjunctival injection or delivery using drug-coated stainless-steel solid MNs)[18,40]. Even though the efficacy of highly-targeted ocular drug delivery using HA-only MN is much better than topical application, such rapid drug delivery could not efficiently suppress the continuous outgrowth of immature blood vessels because of the fast drug clearance by natural fluid circulation in the eye. The same problem makes conventional intraocular injection ineffective for many eye diseases[4,7].

In comparison, we found that DC101 delivered through DL-MN (~1 µg equally divided into the inner core and outer shell of MN) offered much improved therapeutic effect with 90% reduction of neovascular area ($0.12 \pm 0.17$ mm$^2$) (Fig. 7). Both the vessel extension (from the limbus) and circumference (in clock hours) were significantly reduced by DC101-loaded DL-MN patches, owing to the bi-phasic release profile of DC101. DL-MNs not only provide initial bolus dose to quickly reach the therapeutic level at the early onset of the disease, but also sustain drug release to maintain therapeutic effect for a much longer period. Eyes treated with control IgG using either HA-only MN or DL-MN showed no significant changes compared with the untreated ones. In summary, the desirable therapeutic outcome observed in DL-MN treated group is attributable to both highly-targeted and controlled drug delivery.

**Combinational therapy using double-layered MNs for synergistic effect.** Ocular delivery of multiple drugs at different stages of the disease progression can offer a more effective treatment outcome due to their synergistic effects. It is well known that the initial inflammatory response is a key factor to trigger ocular NV (e.g. corneal NV, uveitis-related ocular NV). Under chronic inflammatory condition, inflammatory cells (e.g. macrophages) produce a large number of pro-inflammatory cytokines (e.g. interleukin 6, IL6) and angiogenic growth factors (particularly VEGF), which creates a vicious circle of persistent inflammation and NV. To tackle this issue, DL-MNs were loaded with two

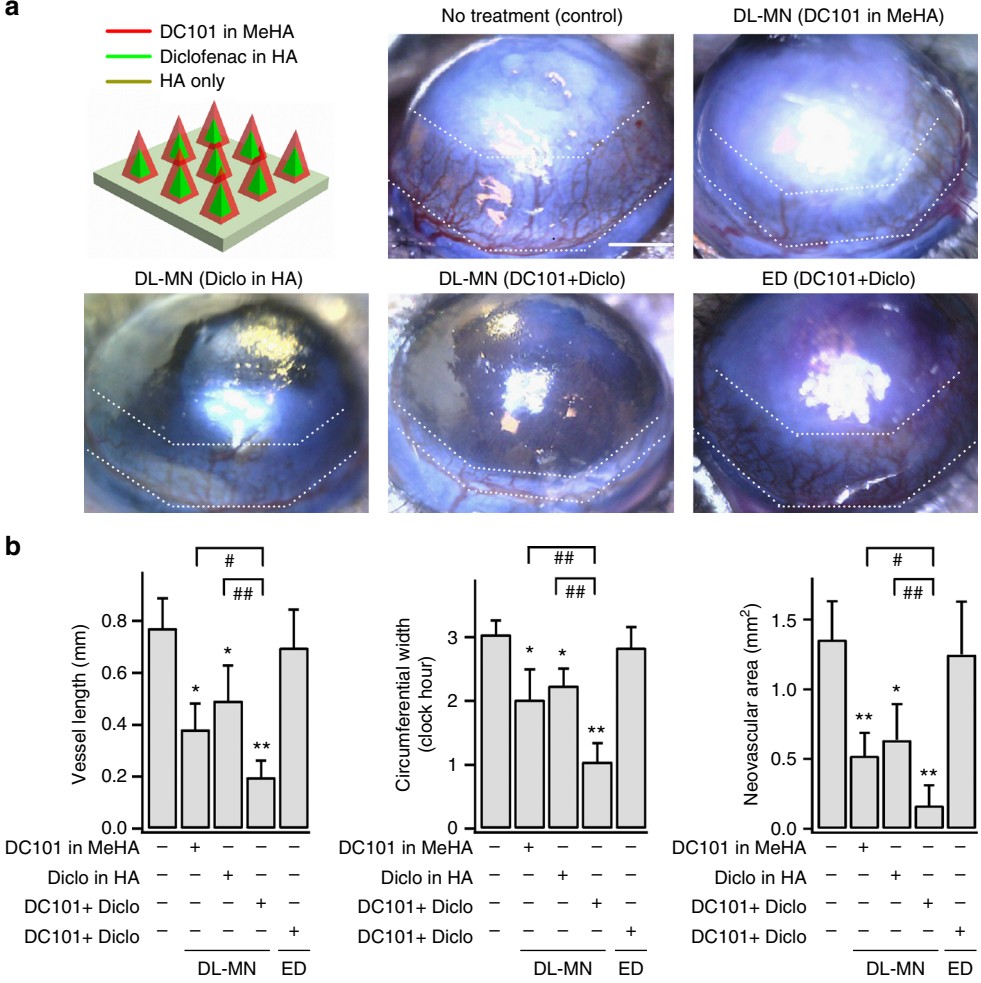

**Fig. 8** Combinational therapy using DL-MN patch for synergistic effect. Mouse eyes were treated differently 2 days after being inflicted with alkali-burn, and examined at day 7. **a** Illustration of drug loadings in DL-MNs, and representative images of differently treated eyes. Scale bar = 500 µm. **b** Quantifications of corneal neovascularization (mean ± SEM; 5 samples for each group). The white dotted lines indicate the extent of neovascular outgrowth from the limbus. Statistical comparison between groups was performed using one way ANOVA. *$p < 0.05$, **$p < 0.01$ vs. control; #$p < 0.05$, ##$p < 0.01$ between indicated pairs

drugs, nonsteroidal anti-inflammatory drug (1 μg diclofenac) in its fast-dissolving HA core and anti-VEGFR2 drug (0.5 μg DC101) in slow-dissolving crosslinked MeHA shell. As shown in Fig. 8, ocular delivery of either only DC101 in MeHA layer or only diclofenac in HA core using DL-MN patch exerted inhibition on neovascular area ($0.52 \pm 0.20$ and $0.63 \pm 0.25$ mm², respectively). The therapy combining both drugs was much more effective ($0.16 \pm 0.24$ mm²). We further show in Supplementary Fig. 6 that that even the high dosage of either one of the drugs, i.e. diclofenac (2 or 5 μg in HA) without DC101 or DC101 (1 or 2.5 μg in MeHA) without diclofenac, is not able to attain the therapeutic outcomes as good as provided by the combinational delivery (1 μg diclofenac in HA plus 0.5 μg DC101 in MeHA). This experiment further confirms the synergistic combination of these two types of drugs. In comparison, topical instillation of same-dosage of both DC101 and diclofenac produced no significant therapeutic effect (Fig. 8).

We further analysed the corneal inflammation by immunofluorescence staining (Fig. 9a, b), and demonstrate that the cornea

treated with diclofenac alone in HA core showed significantly fewer infiltrating F4/80-positive macrophages as compared to the untreated cornea. Although DC101 alone (loaded in MeHA shell) suppressed corneal inflammation to a lesser extent, DL-MNs-loaded with both diclofenac and DC101 showed most significant suppression on the number of infiltrating macrophages. Taken together, these observations show that (i) fast release of diclofenac alone mainly suppresses inflammation with limited anti-angiogenic effects; (ii) conversely, the slow release of DC101 is poorly effective in inhibiting inflammation despite its strong anti-angiogenic effect; (iii) a quick release of diclofenac followed by a sustained release of DC101 led to a much better treatment outcome.

Tear fluid can accurately reflect the dynamic changes of ocular surface tissue (e.g. cornea, sclera)[41]. As MeHA-based patch is highly swellable[24], it was used to collect mouse tear film for analysis of the concentration of inflammatory and angiogenic cytokines (e.g. IL6, VEGF) in tear film after treatment (Fig. 9c). As shown in Fig. 9d, the pore size of the fully swelled MeHA-patch is 2–5 μm, suggesting that large proteins can be easily

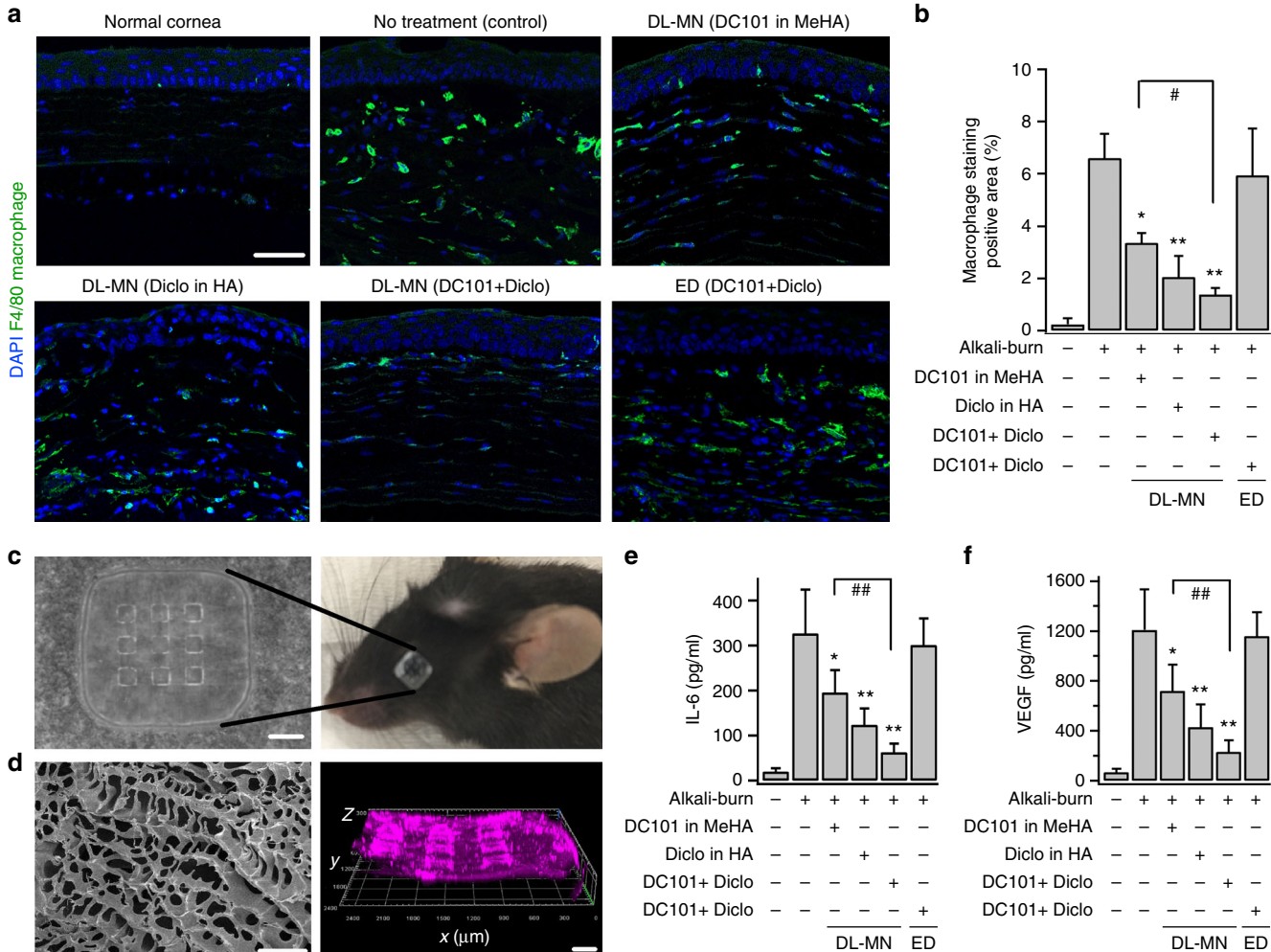

**Fig. 9** Inflammation assessment of combinational therapy using DL-MN patch. Mouse eyes were treated differently 2 days after being inflicted with alkali-burn, and examined at day 7. **a** Immunohistochemical staining of the cornea with a specific macrophage marker—F4/80 surface antigen (green colour). Scale bar = 50 μm. **b** Quantifications of macrophage accumulation (% of positive staining of F4/80; ED: eye drop) (n = 4). **c–f** Cytokine concentrations in collected tear film of differently treated eyes. Tear films were collected with MeHA-based MN-free eye patches (patches were put on the eyes for 1 min to absorb tear film). **c** Scale bar = 500 μm. **d** Scanning electron microscopy image of the patch soaked with PBS (scale bar = 10 μm), and confocal image of the patch with cy5-conjugated albumin absorbed from agarose hydrogel (scale bar = 200 μm). Cytokine concentrations (IL6 and VEGF) were measured with specific ELISA kits (**e**, **f**) (n = 5). Data represents mean ± SEM. n represents the number of samples for each group. Statistical comparison between groups was performed using one way ANOVA. *p < 0.05, **p < 0.01 vs. control (no treatment on alkali-burn eyes); #p < 0.05, ##p < 0.01 between indicated pairs

absorbed. The absorption of Cy5-conjugated albumin from agarose hydrogel confirms the suitability of the patch to collect biomarkers in tear film. Consistent with the previous study[41], both IL6 and VEGF levels in tear film of burn-induced corneal NV were significantly higher than those in normal tear film (Fig. 9e, f). And consistent with our results of macrophage infiltration shown in Fig. 9a, b, DL-MN delivery of DC101 and diclofenac produced most significant effect on reducing those cytokine levels in tear film.

## Discussion

Effective ocular drug delivery for the treatment of vision-threatening diseases (such as glaucoma and neovascular complications, etc.) remains challenging due to the presence of various anatomical and physiological barriers[4]. Herein, we demonstrate an eye patch with micro-drug-reservoirs self-implantable into the ocular surface tissue for controlled drug release. The flexible patch can be readily applied by gentle and brief thumb pressing on the ocular surface, which is as easy as wearing a disposable contact lens without causing discomfort or requiring high skills. As the micro-drug-reservoirs comprise multiple compartments, they allow the release of the same drug with biphasic kinetics or sequentially release of different drugs for synergistic therapy. Comparing to conventional topical eye drop application which requires repeated high-dose instillation and often associates with systemic side-effects, and comparing to painful and risky intraocular injections in clinics, the demonstrated eye patches offer a unique opportunity for patients to conveniently and effectively manage their eye disorders at home.

Here, corneal NV in mice was used as the disease model to demonstrate the effectiveness of our approach. We demonstrate that intra-corneal delivery of DC101 (a monoclonal antibody that blocks VEGFR2) using micro-implants can achieve ~90% reduction of neovascular area with a single treatment of 1 µg dosage. In comparison, eye drop application of DC101 even at a much higher dosage (10 µg) failed to show significant therapeutic effect. In a previous study, systemic intraperitoneal injection of 1 mg DC101 (every 2nd day for 1 week) only led to marginal effect (~20% reduction of neovascular area)[39]. This is expected as the major challenge faced by ocular drug delivery is the limited ability of drug molecules to penetrate the ocular tissue efficiently, due to the presence of ocular barriers. It has been reported that only 0.1% of bevacizumab (a monoclonal antibody against VEGF) in eye drop can reach aqueous humour[42]. It has been shown that eye drop application of 10 µg axitinib (another anti-VEGF agent; daily for 10 days) failed to show significant effect on corneal NV of mice[43]. Clinical studies in human suggested that repeated high-dosage of topical administration is required for the treatment of corneal NV disease. But still only ~41% or ~48% reduction of neovascular area can be achieved using ranibizumab eye drop (~0.2 mg, 4 times daily for 3 weeks)[36] or bevacizumab eye drop (~0.1 mg, 5 times daily for ~3 months)[44], respectively. Based on our mouse experiments and considering that human cornea is ~20 times larger than mice cornea, single treatment with ~20 µg should be effective for human corneal NV using our approach. As the human cornea (~600 µm thick, ~11 mm in diameter) is thicker and larger than mice cornea (~150 µm, ~2.3 mm)[30,45], a larger patch (e.g. 10 × 10 MN array) with longer MNs (e.g. 800 µm) may be suitable for human. With the ability to overcome the ocular surface barriers for localized delivery with high bioavailability, and achieve multi-phasic release kinetics or synergistic therapy from sequential release of multiple drugs, our MN approach can realize low effective dosage and application frequency. This is important to relieve the patient's burden and enhance patient compliance.

Although DL-MNs were used in our experiments, triple compartmentalization can also be fabricated (Supplementary Fig. 7). Here, crosslinked MeHA was chosen as the polymeric matrix for sustained drug release. Prolonged release for 5–6 days is achieved by simply loading the drug molecules in MeHA matrix. Even slower release can be achieved by conjugating the drug molecules with HA molecules to form nanoparticulate conjugates[46]. Two weeks of sustained release can be attained when HA-IgG conjugates are loaded in MeHA matrix (Supplementary Fig. 8). Apart from HA, other biodegradable and biocompatible polymers may also be utilized, for example, FDA-approved synthetic polymer, poly(lactic-co-glycolic acid) (PLGA). PLGA has been used in conventional intraocular implants for sustained ocular drug delivery (e.g. ozurdex, a dexamethasone-loaded PLGA-based intravitreal implant)[47]. By choosing or mixing different PLGA molecules, mechanical properties of MN and drug release kinetics (several days to months) can be readily tailored[14].

Our MN approach promises for other eye diseases as well, for example, delivery of β-adrenergic receptor blockers or prostaglandin analogues for glaucoma[10], corticosteroids (e.g. prednisolone) for anterior uveitis[48], fluconazole for fungal keratitis[49]. It may also be used for intra-corneal delivery of riboflavin to patients with keratoconus, without the need of corneal epithelial scraping and debridement whereby avoiding post-operative pain, infection and permanent damage usually associated with the traditional surgical methods[50]. In summary, the demonstrated MN eye patch, which implants micro-drug-reservoirs for localized, controlled and efficient ocular drug delivery in a convenient, safe and painless manner, provides a cost-effective and home-based solution for many ocular diseases.

## Methods

**Materials**. Sodium hyaluronate (HA) with different molecular weights were purchased from Bloomage Freda Biopharm. IgG, IgGs conjugated with Alexa Fluor dyes, alamarBlue cell viability reagent, Medium 200, low serum growth supplements, Geltrex LDEV-free reduced growth factor basement membrane matrix, bovine serum albumin (BSA), PBS, penicillin–streptomycin and trypsin-EDTA were obtained from ThermoFisher Scientific. VEGFR2 antibodies (GTX14094/DC101, GTX30654, GTX10972) were obtained from GeneTex. Anti-F4/80 (MAB5580) was from R&D Systems. All other reagents and solvents were purchased from Sigma-Aldrich. All reagents were of analytical grade and used without further purification.

**Fabrication of polymeric patches with micro-drug-reservoirs**. MN patches were prepared via simple micromolding method. Briefly, PDMS (Sylgard 184, Dow Corning) micromolds were created by pouring PDMS solution into the custom-designed stainless steel master-molds (Micropoint Technologies, Singapore), which was followed by degassing (10 min in vacuum oven) and curing (70 °C for 2 h). To fabricate eye patches with DL-MN, MeHA aqueous solution (50 mg/ml MeHA together with 0.5 mg/ml Irgacure 2959 in DI water) was casted into the plasma-treated PDMS micromolds through centrifugation (3220$g$, 5 min). After air-drying at room temperature in a fume hood (~12 h), un-modified HA solution (~50 kDa, 200 mg/ml) was applied and centrifuged (805$g$, 5 min) to fill the MN cavities. After air-drying again for ~12 h, HA solution (<10 kDa, 500 mg/ml) was added to produce a robust supporting substrate, and dried overnight. Finally, MN patches were gently peeled off from the micromolds, and exposed to low-intensity ultraviolet light for 3 min (360 nm, ~2 mW per cm²).

**Characterization of MN patches**. MN patches were examined using a field-emission scanning electron microscope (FESEM; JSM-6700, JEOL), and digital microscope (Leica DVM6). MN patches loaded with different IgGs—IgG(405), IgG (488) and IgG(680)—were visualized with a confocal laser scanning microscope (LSM800, Carl Zeiss). The mechanical property of MNs was tested using an Instron 5543 Tensile Tester (Instron). A vertical force was applied to MNs using a flat-headed stainless steel cylindrical probe (at a constant speed of 0.5 mm/min). The force was continuously recorded until a displacement of 450 µm was reached. MN insertion test was performed on the isolated porcine cornea. Briefly, MNs (3 × 3 MN patch, loaded with or without 2 µg IgG) were mounted onto the cylindrical probe, and pressed perpendicular to the cornea at a rate of 5 mm/min until a pre-set maximum load of 4 N was reached. Force exerted on the cornea by the MN as a function of its displacement into cornea was recorded. The insertion force was estimated when the force against the cornea showed discontinuity followed by a

steep slope[27]. The transmittance of the MNs (loaded with or without 2 μg IgG), isolated porcine cornea and aqueous humour (in PBS) was measured in a spectrophotometer (Shimadzu UV-1800).

To evaluate the in vitro biocompatibility of MNs, primary human corneal epithelial cells (Merck Millipore, SCCE016) grown with the EpiGRO ocular complete media (Chemicon, Merck) were exposed to different types of MNs for 2 days, before analysing cell morphology using an inverted microscope (IX71, Olympus, equipped with a digital camera OlympusE330) and cytotoxicity test using an almarBlue cell viability assay. The absorbance of the incubated media containing 10% alamarBlue (~3 h) was measured at 570 nm using a plate reader (SpectraMax M5, Molecular Devices).

The protein bands of IgGs being released from MNs (on 12% polyacrylamide gel) were stained with InstantBlue solution (Expedeon) and imaged with a G:BOX Chemi XT4 imaging system (Syngene). The ability of released IgGs to bind IgG-specific proteins on human umbilical vein endothelial cells (HUVEC; Sigma-Aldrich, 200P-05N) was confirmed by immunofluorescence staining. Briefly, cells were fixed with 4% formaldehyde solution (15 min) before washing with PBS (3 times, 5 min each). After being blocked with 1% BSA in PBST (PBS with 0.1% tween20) (1 h), the cells were incubated overnight in PBST containing 1% BSA and anti-VEGFR2 IgG (free IgG or IgG released from MNs). After washing with PBS, the cells were then incubated with the secondary antibody tagged with Alexa Fluor 680 (2 h), before washing again, and imaging using a confocal microscope.

To evaluate the in vitro bioactivity of IgGs loaded in MNs, HUVECs exposed to 10 ng/ml VEGF (Santa Cruz Biotechnology) were treated with different doses of anti-VEGFR2 IgG (free IgG or IgG released from MNs) for ~18 h. Tube formation of HUVECs grown on the Geltrex matrix (ThermoFisher) was then recorded using an inverted microscope, and tube lengths were measured using the ImageJ (NIH.gov).

To determine the in vitro insertion capability, MN patches (3 × 3 MN array) were applied briefly to the porcine cornea (~30 s), before the supporting base was removed. The corneas were then excised, washed with PBS and analysed for the presence of fluorescence spots produced from IgG(680) loaded in MNs using confocal microscopy. For histological analyses, the corneal tissues were fixed with 4% formaldehyde solution (24 h), and cryoprotected with 30% sucrose solution (24 h), before embedding in FSC22 Frozen Section Media (Leica Microsystem) for cryosectioning (5 μm thick) (CM1950 cryostat, Leica Microsystems), and haematoxylin and eosin staining (Sigma-Aldrich).

**Evaluate in vitro and in vivo drug release profiles**. To evaluate the release profiles of MNs, MNs were immersed in 0.5 ml of simulated tear fluid (NaCl 0.68 g, NaHCO₃ 0.22 g, KCl 0.14 g, CaCl₂·2H₂O 0.008 g, in 100 ml DI water, pH 7.4), gelatin hydrogel (15% w/v gelatin type b from Sigma-Aldrich, in DI water, pH 7.4) or PBS (pH 7.4), and placed in an incubator shaker (50 rpm, 37 °C). IgG molecules (conjugated with Alexa Fluor dyes) released from MNs were measured using a fluorescence spectrometer (SpectraMax M5, Molecular Devices). The real-time visualization of IgG releases from DL-MNs in agarose hydrogel (1.4% w/v, in DI water, pH 7.4) or porcine cornea was analysed using a confocal microscopy.

The in vivo fluorescence imaging of IgG release from MNs was conducted on mice (C57BL/6J, 7–8 weeks old male). Specifically, under anesthetized condition, a MN patch with 3 × 3 array of IgG(680)-loaded DL-MNs was gently applied on one cornea (one eye only) for 30 s. The mice were then imaged immediately (day 0) or at day 3 using an in vivo imaging system (IVIS Spectrum, Perkin Elmer). In some experiments, MN-treated mice were euthanized before collecting the eyeballs and incising the cornea to identify the presence of fluorescence spots produced by MNs using confocal microscopy.

The in vivo bio-distribution of IgG(680) released from MNs was also analysed. Briefly, mice were divided into 4 groups, with the treatment of 10 μg IgG(680) through intraperitoneal injection, topical eye-drop instillation or intra-corneal delivery using MN patch, or without treatment as a control. At 2 h post-treatment, the mice were euthanized and the eyes and the major organs (liver, heart, kidney and lung) were dissected and visualized by IVIS imaging system.

**In vivo studies of MN patches**. All the animal experiments were approved by the Nanyang Technological University, Institutional Animal Care and Use Committee (NTU-IACUC) under protocol ARF-A0350. The mice were housed in light and temperature controlled facility (12-h light/12-h dark cycle, 21 °C), and allowed free access to water and normal diet. DL-MN patches (3 × 3 MN array) were applied onto the central cornea area of anesthetized mice for 30 s. After removing the supporting base, corneas were imaged by a bright-field microscope. After receiving MN insertion, mice were immediately returned to their cages, allowing recovery from anaesthesia. After 10 min, mouse behaviour was monitored to access their pain (Grimace scales: orbital tightening, nose bulge, check bulge and ear position). The body weight and food intake were also recorded (days 1, 3 and 7). In some tested mice groups, mice were euthanized immediately after MN insertion or later (days 1, 3 and 7), and corneas were collected to examine histological changes.

**Ocular burn mouse model for ocular delivery of MN patches**. Chemical-burn injury to the mouse eye was inflicted by placing a sterilized Whatman filter paper (2 mm) soaked with 1 N NaOH solution for 30 s under the anesthetized condition (day 0). Eyes were then extensively flushed with sterilized PBS solution (~10 ml)

using a syringe. At day 2, corneal neovascular outgrowths were imaged by a microscope. Mice were then randomly divided into 6 groups, treated only once with eye-drop instillation of either non-specific control IgG or anti-VEGFR2-IgG (DC101) (10 μg in 20 μl PBS), application of HA-only MN loaded with either control IgG or DC101 (1 μg), or application of DL-MN loaded with either control IgG or DC101 (1 μg, ~0.5 μg each in both layers). For testing the ocular delivery of two drugs, mice were randomly divided into different groups, treated only once with DC101-loaded DL-MN (0.5 μg in MeHA layer), diclofenac-loaded DL-MN (1 μg in HA core), 2-drug-loaded DL-MN (DC101 in MeHA and diclofenac in HA core), or eye-drop instillation of both drugs. For all the interventions, anaesthesia was maintained throughout the procedure with 2% isoflurane. After 5–6 days, corneas were imaged and corneal NV was analysed using ImageJ. Briefly, the vessel lengths (VL) were measured from the limbus to their leading edges, sectoral circumference was expressed as clock hours (CH, 1 being 30 degrees of arc), and the vessel area (VA) was calculated using the formula, $VA = 0.2\pi \times VL \times CH$[51]. In some experiments, mice were euthanized and corneas were collected for immunohistochemistry analyses of macrophage infiltration in corneal stromal layer. F4/80 as a major macrophage marker was stained with a specific antibody. Tear films from differently treated mice were collected by simply placing MeHA patch without MNs on their ocular surface for 1 min, before centrifugation of the patch at 16,000g for 5 min. Subsequently, interleukin-6 (IL6) and VEGF concentrations were determined by IL6 and VEGF ELISA kits (Invitrogen).

**Statistical analyses**. Quantitative data were represented as mean ± SEM. Statistical analysis was performed using one-way analysis of variance (ANOVA) followed by Tukey's post-hoc test. A p value of <0.05 was considered to be statistically significant.

## Data availability
All relevant data are included in the main manuscript and the Supplementary Information. Additional data are available from the corresponding authors upon reasonable request.

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

## Acknowledgements

This research was supported by Singapore National Research Foundation under its CBRG grants (NMRC/CBRG/0070/2014 and NMRC/CBRG/0058/2014) administered by the Singapore Ministry of Health's National Medical Research Council; Singapore Ministry of Education under its AcRF Tier 1 Grants RG126/15, RG 131/15 and RG52/17-(S) and AcRF tier 2 grant (MOE2017-T2-2-005); Singapore A*STAR Biomedical Research Council under its IAF-PP grant.

## Author contributions

A.T., C.L., H.C. and P.K.D. performed research. A.T., H.C. and P.K.D. fabricated microneedles. A.T., C.L. and P.K.D. conducted animal experiments. A.T. and P.C. planned the experiments, analysed the results and drafted the manuscript. C.M.G.C. contributed discussion and provided clinical insights. P.C., X.W. and C.X. designed and supervised the studies, and finalized the manuscript. All authors reviewed and approved the final version of the manuscript.

## Additional information

**Competing interests:** The authors declare no competing interests.

