## [Peer Review File · Nature Communications]

Reviewers' comments:

Reviewer #1 (Remarks to the Author):

This manuscript describes development of double-layered detachable microneedles for corneal drug delivery. The key idea is the use of double layers comprised of HA and crosslinked MeHA for fast and slow drug release after anchored in the cornea. Use of dual drug delivery for detachable microneedles for ocular drug delivery are certainly desired in the field. The concept and the results are well described. However, there are still couple of questions that need clarification before decision for publication.

<major issues>

1. The detachability is a key issue for detachable microneedles. It seems that the authors relied on the swelling and softening of HA for detachability. However, the characterization and success probability are not clearly presented.

2. Detachment of microneedles is not clear in the Fig. 4 images. How did you determine which microneedles were detached from the substrate.

3. Although the authors argued that microneedles were inserted (or embedded) in the cornea tissue, how did you confirm it? Can they be attached to the surface of the cornea? Some microneedles make insertion and they retract from their insertion spots due to elasticity of the tissue.

4. Observation of cornea tissue architecture in Fig, 5f is meaningless, if the authors used different animals at each time point. I presume the animals were sacrificed at each time point. Separation of stromal lamella often occurs during paraffin fixation and microtome process. In general, explanations given in the lines 201- 222 are vague and hard to understand.

5. Although the combinatorial delivery seems good, I guess increase of diclofenac without DC-101 may result in the same result.

<minor issues>

1. At the bottom of page 5 and the beginning of page 6, the authors mentioned that Me-HA microneedles were weaker than HA microneedles. This doesn't make sense. Isn't Me-HA microneedles crosslinked?

2. Compression test is not so accurate for mechanical characterization of microneedles, since most polymer tips fails due to tip collapse. Measurement of microneedle insertion force into the tissue sample is more accurate to understand microneedle behaviour.

Reviewer #2 (Remarks to the Author):

The manuscript details a new strategy for drug delivery using an eye patch equipped with an array of detachable microneedles (MNs). This is particularly an important study as several MN based strategies have been developed so far for efficient ocular delivery, but no such patient-friendly, home-based treatment has been established. While the study is performed well in majority, some clarifications are required.

1. Pg 2: Introduction: Advantages of MNs over self-implantable contact lenses are not discussed. The resurgence of contact lenses in offering non-invasive, sustained drug delivery with minimal burst effect has tremendously improved ocular drug delivery. For example, Alvarez-Lorenzo et al. have reported norfloxacin-imprinted lenses released only 40% of the drug in 24 h with less than

10% burst release during the first 2 h.

2. Pg 2: Line 64: Topical administration and local injection only produce burst release....This statement needs references. Burst release can possibly be prevented with the advancement in hydrogel applications.

3. "MNs can penetrate the ocular barriers (epithelial and stromal layers of cornea) with minimal invasiveness and be self-implanted as drug reservoirs for controlled drug release". Statement needs reference.

4. Pg 4: Paragraph 2: Bioactivity of released IgGs.....It is not clear which samples (day/time period) were utilized for the bioactivity studies. Fig 3A suggests IgG release was prolonged up to 120 hr. Authors ought to report the bioactivity of IgG released at 120 hr.

5. Pg 4: Results: Rationale for using PBS for release studies ought to be provided. Authors should conduct the release studies in conditions that closely relate to the ocular conditions.

6. Pg 4: Results: Paragraph 4: in-vivo release profile....This is a very curious finding. However, if the in-vivo release data can be quantified and correlated with the in-vitro drug release, a relationship/correlation can be developed to prove such a release behavior.

7. Pg 4: Line 173: MNs are essentially transparent...this statement needs to be justified with an experiment. The clarity/transparency of MNs can be compared with the aqueous humor or transmittance can be determined and compared.

8. Pg 7: Paragraph 1: Fig. 6A needs to be explained well in order to justify the therapeutic effect of DL-MNs (regression of white region or any other related changes that could be inferred).

9. Pg 7: Combinational therapy: Diclofenac is a hydrophobic drug. Does the results indicate that the HA core can be used to entrap both hydrophilic and hydrophobic drugs? Can this be accomplished in the other way i.e. DC101 in HA core and Diclofenac in MeHA layer? It would be crucial to determine the therapeutic effect of such a combination. In addition, authors ought to report the amount of diclofenac entrapped in the HA core.

10. The figures lack consistency in the labelling of treatment groups. Fig. 7 a & b ought to be labelled the same way. Fig 7b labels should be modified to DL-MN (DC101 in MeHA), DL-MN (Diclo in HA) and DL-MN (DC101+Diclo). Similarly, Fig. 8 e and f labels ought to be corrected.

11. Supplementary Figure 1b: Scale should be same as Supp Fig 1c. Proton signals at 6.1 ppm cannot be visualized in Supp Fig 1b.

12. It is more appropriate to use "sustained" delivery than "controlled" delivery throughout the manuscript. It is highly unlikely to control the release of drugs/agents from a nanocarrier until and unless the release is mediated by any kind of stimuli.

13. Pg 10: Line 445: agarose spelling should be corrected.

14. Please cite these two papers: "Microneedles: A New Frontier in Nanomedicine Delivery" and "Ocular delivery of proteins and peptides: Challenges and novel formulation approaches" which discuss in detail the latest technologies and issues related to ocular drug delivery.

Reviewer #3 (Remarks to the Author):

This paper proposed dissolvable double layer microneedles (MN) from Hyaluronic acid (HA) and crosslinked MeHA for ocular drug delivery. Authors demonstrated that the MN produce biphasic drug release which is beneficial for many types of treatments. The performance of the MN is then successfully tested in the pig cornea as well as in in vivo mice model. The work is novel and addresses some of the current limitations and challenges of using MN for ocular drug delivery.

Specific Comments:

Page 3. The rationale of making MeHA, a softer material, an outside layer of the MN is not obvious. Is it possible to manufacture MN with dissolvable HA on the outside surface? Could this further speed the burst release?

Page 4. MN were injected in the agarose hydrogel to visualise drug release, but the drug diffusion rate in the agarose is likely much faster than in the cornea tissue. Is it feasible to do similar

experiments in pig cornea tissue?

Page 5. Some comment and discussion is needed when relating mice in vivo experiments to potential human application. Authors mention that pig cornea is more suitable, so how mice cornea compares to human and pig one? Say, if it is thinner will it necessitate longer MN in humans to have comparable results?

Page 9. Please give g values instead rpm here and further in the text.

Page 10. Please give details of PBS release experiment: was the solution stirred? What was pH? Receptor volume?

Reviewer #1:

This manuscript describes development of double-layered detachable microneedles for corneal drug delivery. The key idea is the use of double layers comprised of HA and crosslinked MeHA for fast and slow drug release after anchored in the cornea. Use of dual drug delivery for detachable microneedles for ocular drug delivery are certainly desired in the field. The concept and the results are well described. However, there are still couple of questions that need clarification before decision for publication.

Comment 1: The detachability is a key issue for detachable microneedles. It seems that the authors relied on the swelling and softening of HA for detachability. However, the characterization and success probability are not clearly presented.

Answer: We thank the reviewer for the insightful comment. The supporting substrate is made of HA of very low molecular weight (miniHA, 3-10 kDa) which has a very fast dissolution rate (Small Methods 2017, 1700269). As soon as MNs penetrate into the cornea, the tissue fluid and surface fluid are quickly drawn into the hydrophilic HA matrix and dissolve the miniHA molecules at MN-substrate junction. In addition, the mechanic perturbation caused by the penetration and substrate removal process also facilitates MN detachment. When being applied to agarose hydrogel or mouse cornea or porcine cornea, the embedment percentage of MNs is 100%, ~90%, or ~85% respectively as evidenced by visual inspection of the removed substrate (Fig. 5c and d, Fig. 6b) and embedded MNs (Fig.4b, 5d), and the fluorescence marks left behind (Fig. 5g, Fig. 6e). The above discussion has now been added in the manuscript.

Comment 2: Detachment of microneedles is not clear in the Fig. 4 images. How did you determine which microneedles were detached from the substrate.

Answer: Fig. 4b (0 min – 6 hr) shows the confocal visualization of agarose hydrogel with DL-MNs detached from the supporting substrate. In the revised Fig. 4b, we now show at first an image of DL-MNs with the supporting substrate for comparison. Please refer to our answer above regarding the characterization and success probability.

Comment 3: Although the authors argued that microneedles were inserted (or embedded) in the cornea tissue, how did you confirm it? Can they be attached to the surface of the cornea? Some microneedles make insertion and they retract from their insertion spots due to elasticity of the tissue.

Answer: Embedment of MNs is confirmed by visual inspection of the removed substrate (Fig. 5c and d, Fig. 6b) and embedded MNs (Fig.4b, 5d), the fluorescence marks left behind (Fig. 5g, Fig. 6e), as well as the observed insertion cavities (Fig.5e, 6f). In our experiments, the eyes are flushed with PBS after MN insertion. If MNs are just attached to the surface of the cornea, they would be rinsed away. In addition, if they are just attached on the surface, we would not be able to observe good therapeutic effects.

Comment 4: Observation of cornea tissue architecture in Fig, 5f is meaningless, if the authors used different animals at each time point. I presume the animals were sacrificed at each time point. Separation of stromal lamella often occurs during paraffin fixation and microtome process. In general, explanations given in the lines 201- 222 are vague and hard to understand.

Answer: It is not possible to observe histological changes of cornea in the same mouse. But our observation from different mice at day0, day1, day3 or day7 is consistent. Specifically, the insertion cavities are present in all mice at day0, and become absent in all mice at day3 and day7. Fig. 6g shows the representative images at different days to illustrate the trend. We think this information is important to show the invasiveness of our method. In addition, this figure shows the insertion cavity reflecting the MN penetration depth and width. We have accordingly revised the description in the manuscript to make it clearer.

We agree with the reviewer that 'Separation of stromal lamella often occurs during paraffin fixation and microtome process.' Therefore, this sentence is now removed.

Comment 5: Although the combinatorial delivery seems good, I guess increase of diclofenac without DC-101 may result in the same result.

Answer: We thank the reviewer for the good suggestion. Accordingly, we have conducted a new experiment, and found that even the high dosage of either one of the drugs alone, i.e., diclofenac (2 µg or 5 µg in HA) without DC101 or DC101 (1 µg or 2.5 µg in MeHA) without diclofenac, is not able to attain the therapeutic outcomes as good as provided by the combinational delivery (1 µg Diclofenac in HA plus 0.5 µg DC101 in MeHA). This experiment further confirms the synergistic combination of these two types of drugs. A new Figure (Supplemental Fig.6 in SI) and corresponding description / discussion have been added.

Comment 6: At the bottom of page 5 and the beginning of page 6, the authors mentioned that Me-HA microneedles were weaker than HA microneedles. This doesn't make sense. Isn't Me-HA microneedles crosslinked?

Answer: Cross-linking doesn't necessarily enhance the mechanical strength. MeHA MN is mechanically weaker than HA MN (Fig. 3h) is mainly because MeHA MN is made of a lesser amount of polymer molecules. As the high molecular weight MeHA (~300 kDa) is needed in order to make it highly swellable and capable of sustained drug release, its solution is highly viscous. As the consequence, we can only use a relatively low concentration (50 mg/ml) to fabricate MNs. In comparison, a high concentration HA solution (200 mg/ml) is used to fabricate HA MNs. The explanation is now provided in the manuscript.

Comment 7: Compression test is not so accurate for mechanical characterization of microneedles, since most polymer tips fail due to tip collapse. Measurement of microneedle insertion force into the tissue sample is more accurate to understand microneedle behaviour.

Answer: We thank the reviewer for the good suggestion. Following the suggestion, a new experiment has been conducted to measure insertion force of MNs into the cornea isolated from pig eyes (see new Fig. 5f). Description and discussion of this new experiment are added accordingly.

Reviewer #2:

The manuscript details a new strategy for drug delivery using an eye patch equipped with an array of detachable microneedles (MNs). This is particularly an important study as several MN based strategies have been developed so far for efficient ocular delivery, but no such patient-friendly, home-

based treatment has been established. While the study is performed well in majority, some clarifications are required.

Comment 1: Pg 2: Introduction: Advantages of MNs over self-implantable contact lenses are not discussed. The resurgence of contact lenses in offering non-invasive, sustained drug delivery with minimal burst effect has tremendously improved ocular drug delivery. For example, Alvarez-Lorenzo et al. have reported norfloxacin-imprinted lenses released only 40% of the drug in 24 h with less than 10% burst release during the first 2 h.

Answer: We thanks reviewer for the good suggestion. We have now added a discussion on drug-loaded contact lens for ocular drug delivery in the introduction, with two related references cited.

Comment 2: Pg 2: Line 64: Topical administration and local injection only produce burst release....This statement needs references. Burst release can possibly be prevented with the advancement in hydrogel applications.

Answer: We agree with the reviewer. Accordingly, “conventional” is added before “topical administration and local injection” and two review articles on ocular drug delivery are cited.

Comment 3: “MNs can penetrate the ocular barriers (epithelial and stromal layers of cornea) with minimal invasiveness and be self-implanted as drug reservoirs for controlled drug release”. Statement needs reference.

Answer: That statement is a short description of our results. To the best of our knowledge, there is no publication on this so far.

Comment 4: Pg 4: Paragraph 2: Bioactivity of released IgGs.....It is not clear which samples (day/time period) were utilized for the bioactivity studies. Fig 3A suggests IgG release was prolonged up to 120 hr. Authors ought to report the bioactivity of IgG released at 120 hr.

Answer: We thank the reviewer for the careful reading. To make it clear, we have revised the description in the manuscript as well as in the legends for Supplementary Fig. 3c and Fig. 4a. The bioactivity of IgG released at 120 hr is shown in Supplementary Fig. 4a, proving that IgG stored in MNs for 5 days and remained in solution for another 5 days (120 hrs) is still bioactive.

Comment 5: Pg 4: Results: Rationale for using PBS for release studies ought to be provided. Authors should conduct the release studies in conditions that closely relate to the ocular conditions.

Answer: We appreciate the reviewer’s valuable suggestion. Following the suggestion, we have conducted additional release studies using simulated tear fluid and 15% gelatin hydrogel. The results are presented in the revised Fig. 3f. Simulated tear fluid is widely used to simulate ocular drug delivery. And we chose gelatin hydrogel to mimic the corneal stromal tissue (which consists of 15-16% of collagen and ~80% of water) as many studies suggest that gelatin hydrogel is suitable to corneal tissue engineering (Zhi Chen et al 2018 Biomed. Mater. 13 032002). The corresponding description of experiments and discussion are added in the manuscript.

Comment 6: Pg 4: Results: Paragraph 4: in-vivo release profile....This is a very curious finding. However, if the in-vivo release data can be quantified and correlated with the in-vitro drug release, a relationship/correlation can be developed to prove such a release behavior.

Answer: We thank the reviewer for the good suggestion. To quantify the biphasic release shown in Fig. 4, we have measured the fluorescence intensity change of IgGs released from MNs into the agarose gel (see the new Supplementary Fig. 5a).

In addition, we have conducted a new experiment using porcine cornea. The real-time visualization of IgG release from DL-MNs in porcine cornea, and the corresponding time course of the fluorescence intensity change at the area adjacent to DL-MNs are shown in the new Figure 5g-i.

Taken together the results shown in both agarose hydrogel and porcine cornea, we demonstrate the biphasic release kinetics of DL-MNs.

Comment 7: Pg 4: Line 173: MNs are essentially transparent...this statement needs to be justified with an experiment. The clarity/transparency of MNs can be compared with the aqueous humor or transmittance can be determined and compared.

Answer: We thanks reviewer for the suggestion. A new experiment has been conducted to measure the transparency of MNs, and compare with cornea and aqueous humor isolated from porcine eye (new Figure 5i). And we observed that “the transmittance of fully hydrated DL-MNs within the visible range is about 73-86%, which is comparable to that of cornea and aqueous humour”.

Comment 8: Pg 7: Paragraph 1: Fig. 6A needs to be explained well in order to justify the therapeutic effect of DL-MNs (regression of white region or any other related changes that could be inferred).

Answer: We thank the reviewer for the careful reading. But actually, the white regions in the original Fig. 6a (now Fig. 7a) are just the reflection of light. To make the figure clearer, two dotted lines are drawn to indicate the extent of neovascular outgrowth from the limbus. This figure is explained in details in the manuscript as follows: “Similar to the un-treated eyes, eyes treated with control IgG eye-drop showed substantial corneal NV (the blood vessel outgrowth from limbus) ($1.48 \pm 0.45 \text{ mm}^2$ vs. $1.50 \pm 0.24 \text{ mm}^2$) (day 7). Similarly, topical delivery of DC101 via eye-drop (with a high dose of $10 \mu\text{g}$ in $10 \mu\text{l}$) had no significant effect on corneal NV as compared to the untreated eyes ($1.24 \pm 0.21 \text{ mm}^2$). In contrast, eyes treated with DC101 ($\sim 1 \mu\text{g}$) delivered through fast-dissolving HA-only MNs led to $\sim 44\%$ reduction in neovascular area ($0.84 \pm 0.43 \text{ mm}^2$) (Fig.7).....In comparison, we found that DC101 delivered through DL-MN ($\sim 1 \mu\text{g}$ equally divided into the inner core and outer shell of MN) offered much improved therapeutic effect with 90% reduction of neovascular area ($0.12 \pm 0.17 \text{ mm}^2$) (Fig.7).”

Comment 9: Pg 7: Combinational therapy: Diclofenac is a hydrophobic drug. Does the results indicate that the HA core can used to entrap both hydrophilic and hydrophobic drugs? Can this be accomplished in the other way i.e. DC101 in HA core and Diclofenac in MeHA layer? It would be crucial to determine the therapeutic effect of such a combination. In addition, authors ought to report the amount of diclofenac entrapped in the HA core.

Answer: We thank the reviewer for the insightful comments. Indeed, HA can be used to entrap both hydrophilic and hydrophobic drugs. This has also been previously demonstrated in other studies. We have now highlighted this advantage in the manuscript.

We used corneal neovascularization as our disease model. For this disease, the initial inflammatory response is the key factor to trigger ocular neovascularization. Therefore, we load Diclofenac (anti-inflammatory drug) in HA core in order to give the initial fast release, while DC101 (anti-angiogenic drug) is loaded in MeHA layer in order to achieve sustained release to suppress the disease progression. Therefore, DC101 in HA core and Diclofenac in MeHA layer will not work effectively. As shown in Fig.7, the fast release of DC101 cannot provide good therapeutic effect.

Amount of diclofenac entrapped in the HA core has stated in the Result section. "DL-MNs were loaded with two drugs, nonsteroidal anti-inflammatory drug- (1 µg diclofenac) in its fast-dissolving HA core and anti-VEGFR2 drug (0.5 µg DC101) in slow-dissolving crosslinked MeHA shell."

Comment 10: The figures lacks consistency in the labelling of treatment groups. Fig.7 a & b ought to be labelled the same way. Fig 7b labels should be modified to DL-MN (DC101 in MeHA), DL-MN (Diclo in HA) and DL-MN (DC101+Diclo). Similarly, Fig. 8 e and f labels ought to be corrected.

Answer: We thank reviewer for the careful reading. The figure labelling has now been corrected as suggested by the reviewer.

Comment 11: Supplementary Figure 1b: Scale should be same as Supp Fig 1c. Proton signals at 6.1 ppm cannot be visualized in Supp Fig 1b.

Answer: We thank reviewer for the careful reading. Now we have made the scale the same in these two figures. Proton signal is small but is identifiable.

Comment 12: It is more appropriate to use "sustained" delivery than "controlled" delivery throughout the manuscript. It is highly unlikely to control the release of drugs/agents from a nanocarrier until and unless the release is mediated by any kind of stimuli.

Answer: Through specific designed on both polymer composition and microneedle structure, biphasic release with desired kinetics is achieved. Therefore, it could be regarded as "controlled" delivery. "Sustained" delivery only reflects to the release from MeHA layer, but not the entire picture of our invention.

Comment 13: Pg 10: Line 445: agarose spelling should be corrected.

Answer: We thank the reviewer for the careful reading. The spelling is now corrected.

Comment 14: Please cite these two papers: "Microneedles: A New Frontier in Nanomedicine Delivery" and "Ocular delivery of proteins and peptides: Challenges and novel formulation approaches" which discusses in detail the latest technologies and issues related to ocular drug delivery.

Answer: We agree with the suggestion. These two relevant papers have now been cited.

Reviewer #3:

This paper proposed dissolvable double layer microneedles (MN) from Hyaluronic acid (HA) and crosslinked MeHA for ocular drug delivery. Authors demonstrated that the MN produce biphasic drug release which is beneficial for many types of treatments. The performance of the MN is then successfully tested in the pig cornea as well as in in vivo mice model. The work is novel and addresses some of the current limitations and challenges of using MN for ocular drug delivery.

Comment 1: Page 3. The rationale of making MeHA, a softer material, an outside layer of the MN is not obvious. Is it possible to manufacture MN with dissolvable HA on the outside surface? Could this further speed the burst release?

Answer: We thank the reviewer for the insightful comments. As we discussed in the main text that “because of the fast dissolving nature of HA, HA-MNs cannot maintain its sharp-pointed structural integrity and mechanical strength during penetration into a wet surface like cornea” and “crosslinked HA is more resistive to dissolution”. Therefore, MeHA was used for the outer layer of MN in our experiments to ensure penetration. To make this point more obvious, we have now added a sentence as follows: “Because the highly dissolvable HA is covered by MeHA, the MNs are able to penetrate the wet cornea surface.”

In addition, the burst release of “HA inner core” is already fast enough (>80% was released within 5 min in the simulated tears and agarose hydrogel, and 30 min in the gelatin hydrogel and corneas) (Fig.3f, 4b, 5i). In our opinion, there is no need to further speed up the burst.

Comment 2: Page 4. MN were injected in the agarose hydrogel to visualise drug release, but the drug diffusion rate in the agarose is likely much faster than in the cornea tissue. Is it feasible to do similar experiments in pig cornea tissue?

Answer: We thank the reviewer for the good suggestion. Accordingly, we have conducted a new experiment in pig cornea tissue.

The real-time visualization of IgG releases from DL-MNs in porcine cornea, and the corresponding fluorescence intensity changes of cornea in the region adjacent to DL-MNs were shown in the new Figure 5g-i. As expected by the reviewer, IgG diffusion rate in the cornea tissue is slower than those in the agarose hydrogel (Fig.4; Supplemental Fig.5a). This new result further confirms the biphasic release kinetics of DL-MNs. The corresponding experiment description and discussion are added in the manuscript.

Comment 3: Page 5. Some comment and discussion is needed when relating mice in vivo experiments to potential human application. Authors mention that pig cornea is more suitable, so how mice cornea compares to human and pig one? Say, if it is thinner will it necessitate longer MN in humans to have comparable results?

Answer: As suggested by the reviewer, we have discussed about potential application of the eye patches on human in the discussion section of revised manuscript, “Based on our mouse experiments and considering that human cornea is ~20 times larger than mice cornea, single treatment with ~20 µg should be effective for human corneal NV using our approach. As the human cornea (~600 µm thick, ~11 mm in diameter) is thicker and larger than mice cornea (~150 µm, ~2.3 mm), a larger patch (e.g., 10 x 10 MN array) with longer MNs (e.g., 800 µm) may be suitable for human.”

Comment 4: Page 9. Please give g values instead rpm here and further in the text.

Answer: As suggested by the reviewer, 'rpm' values have been replaced with 'g' values in the revised manuscript.

Comment 5: Page 10. Please give details of PBS release experiment: was the solution stirred? What was pH? Receptor volume?

Answer: We thank the reviewer for the careful reading. Following the suggestion, we have now elaborated these experimental details.

REVIEWERS' COMMENTS:

Reviewer #1 (Remarks to the Author):

In the previous submission, there were some ambiguity of MN detachability and related characterizations. In this revised version, my comments were answered with clarity and supporting data. I would like to recommend the manuscript for publication.

Reviewer #2 (Remarks to the Author):

The authors have done a great job in revising the manuscript.

REVIEWERS' COMMENTS:

Reviewer #1 (Remarks to the Author):

In the previous submission, there were some ambiguity of MN detachability and related characterizations. In this revised version, my comments were answered with clarity and supporting data. I would like to recommend the manuscript for publication.

Answer: We thank the reviewer for positive comment and recommendation for publication in Nature communications.

Reviewer #2 (Remarks to the Author):

The authors have done a great job in revising the manuscript.

Answer: We thank the reviewer for positive comment.